# Research on the Siting of Rural Public Cultural Space Based on the Path-Clustering Algorithm: A Case Study of Yumin Township, Yushu City, Jilin Province, China

**Deli Liu** * 🆔 **and Keqi Wang** *

School of Architectural and Urban Planing, Jilin Jianzhu University, Changchun 130118, China
* Correspondence: liudeli@student.jlju.edu.cn (D.L.); wangkeqi@jlju.edu.cn (K.W.)

**Abstract:** Cultural revitalization is the foundation of rural revitalization. Rural public cultural spaces are important places for rural residents to participate in cultural activities and important carriers for the occurrence, inheritance and development of rural culture. In this study, we propose a siting analysis method for public cultural spaces based on a path-clustering algorithm. Then, Yumin Township in Jilin Province is used as a case study to test the method. Within the township area, public cultural spaces are divided into three types based on the difference in service scope: "single-village decentralized", "single-village concentration" and "multi-village concentration". This paper focuses on the analysis of the siting of the "multi-village concentration" type of public cultural spaces using a path-clustering algorithm. First, the path-clustering algorithm is written, and then its parameters, initial clustering centers and number of clusters are analyzed for their effect on the clustering results. Two parameters, within-cluster sum of path distance (WCSPD) and maximum value of the path distance within-cluster (MPDWC), are proposed to evaluate the results of the algorithm. Then, taking Yumin Township of Jilin Province as an example, the villages in the township are clustered with travel distance as the main influencing factor. It is ensured that the shortest path distance from the central village in each cluster to other villages is no greater than the set travel distance. Finally, the public cultural space is constructed in the central village to serve all villages in the cluster. This siting analysis method can output a variety of results that can be used as a reference for decision making in rural public cultural space planning.

**Keywords:** cultural revitalization; rural public cultural space; path-clustering algorithm; site

## 1. Introduction

### 1.1. Background

At present, most cities in China have entered the stage of stock development, while the development of the countryside is somewhat lagging behind. Under the background of rural revitalization, an increasing number of people from different fields have started to devote themselves to the construction of the countryside. However, many problems have emerged in these rural development efforts, such as ignoring the individuality of the villages, not respecting the history, culture and natural conditions, and adopting a uniform model to deal with different villages. The reason for these problems is the neglect of rural culture. The essence of Chinese culture is vernacular culture. The roots of Chinese culture are in the countryside [1]. Therefore, in the reconstruction of villages, it is necessary to focus on strengthening the construction of rural culture. The revitalization of rural culture is an inevitable requirement to realize rural revitalization. The production and development of culture cannot be achieved without space as the material carrier. Rural public cultural space is an important place for the occurrence and transmission of rural culture, which carries the spiritual life of farmers. It is not only an important element in strengthening rural cultural construction, but also an important physical carrier for rural cultural revitalization [2].

Over time, an increasing number of scholars in China are paying attention to the important role of public cultural space in rural cultural construction work. Zhang et al. [3] proposed that in the context of rural revitalization, the construction of rural public cultural space should meet the needs of rural residents for a better life. Chen et al. [4] analyzed the impact of accelerated urbanization on rural public cultural space in China and studied the methods and models for reconstructing rural public cultural space. Ma [5] proposed cultivating traditional public cultural spaces in villages into new public cultural spaces by means of transformation or innovation. However, not only do these studies lack scientific and quantitative analysis methods, but most of them only develop some theoretical guidance strategies, which cannot be implemented at the practical level.

*1.2. Related Works*

1.2.1. Rural Public Space

Rural public space, as an important field for the generation of social order in rural transformation [6,7], carries long-standing cultural traditions, historical memories and villagers' nostalgia of the village. It embodies villagers' cultural concepts, ideological beliefs and value pursuits [8]. Rural public space has emerged as a research hotspot in recent years. Many researchers from the fields of geography, sociology, architecture and management have performed much research on the value of rural public space from different research perspectives and space-time scales [9,10]. These include the transformation and reconstruction of rural public space [11,12], the vitality of rural public space [13,14], the relationship between rural public space and rural tourism [15], the relationship between rural public space and population migration and aging [16,17] and other aspects of research.

Public space is the connective structure and ultimate foundation for all human spaces [18]. Cultural space is a collective of many types of spaces that have a profound relationship with people's lifestyles [19]. The public cultural space that we focus on in this paper can be seen as a type of public space. However, the difference is that it has not only spatial properties. It is more of a collection of the pattern of space, the environmental characteristics and above all the surrounding people [19,20]. In this paper, we focus more on the spatial properties of the public cultural space. Therefore, it can be considered a kind of rural public space. The above summary of research on rural public space reveals that there is a need for further research on the analysis methods that can give specific guidance for the design or planning of rural public space. Therefore, in this paper, we propose a path-clustering algorithm and apply it to the study of siting rural public cultural spaces. It provides a reference for the planning of public cultural space in rural areas.

1.2.2. Distribution of Public Facilities

Many scholars have conducted studies on the distribution of public service facilities. The main research directions can be summarized into two. One is to study the distribution of facilities based on the problems that exist in the current situation and propose optimization strategies or recommendations. For example, the distribution of public service space is rationalized and optimized to enhance efficiency. Wang et al. [21] took Wuhan city as an example and tried to use machine learning algorithms for the planning and siting of elderly facilities based on POI data. The purpose is to make the overall layout of elderly facilities better and the resource allocation more reasonable. Sevtsuk et al. [22] introduce a version of the Huff retail expenditure model, where retail demand depends on households' access to retail centers. The study examines the issue of merchandize retail centers in cities and explores how adjusting the location and size of commercial centers can maximize overall visitation. Aliniai et al. [23] conducted a study on the mismatch between the development of cities and public parking lots. The study aims to find the most suitable sites for public parking lots in one of the most crowded cities of western Iran. Wang et al. [24] studied the existing spatial distribution of various types of medical facilities. Based on the current spatial distribution of existing medical facilities, an optimization strategy is proposed to develop an emergency response plan for health emergencies.

Another is the siting planning of new public facilities or spaces based on specific needs. The object of study can be basic energy facilities. Bojić et al. [25] studied the location of the power plant in the province of Vojvodina according to the minimum cost of electricity generation. Home-Ortiz et al. [26] proposed a mixed integer conic programming (MICP) model to find the optimal size, type and location of distributed generators (DGs) within a multistage planning horizon in radial distribution systems. The research object can also be emergency facilities. Shavarani [27] proposed a hybrid genetic algorithm to find the optimal topology for rescue centers and resupply stations. This allows these facilities to cover a large-scale area without exceeding the flight distance of unmanned aerial vehicles (UAVs). Xu et al. [28] developed a scenario-based hybrid bilevel model. The results obtained from this model can be used as a reference for balancing the interests of government and residents in shelter planning in Beijing. These site planning studies of such space facilities are based on the existing regular experience or some objective conditions. For example, the distribution of rescue facilities is set according to the maximum flight distance of the UAV [27]. In this paper, we take Yumin Township as an example to study the siting of public cultural spaces. The conclusions obtained can be used as a reference for the planning of new public cultural spaces. Similar to many other studies [29–31], this article examines the spatial distribution of public cultural spaces based on the accessibility of space. There are three general approaches used to measure spatial access to public spaces [32]: the spatial proximity approach, the container approach and the coverage approach. In the study of this paper, we refer to two approaches, the spatial proximity approach and the coverage approach, to complete the site analysis of public cultural spaces with the preferred travel distance of people as the main influencing factor. The spatial proximity approach mainly considers the cost of travel from the demand population to the space without considering the size and facilities of this space. The coverage approach mainly measures the demand and supply ratio of a public space. Finally, it aims to propose a set of reasonable and effective methods for siting rural public cultural spaces to achieve a reasonable distribution of cultural space resources.

### 1.2.3. Facility Location Problems

The facility siting problem is to locate a set of facilities (resources) in such a way as to minimize the cost of meeting the (customer) demand under certain constraints. The study of location theory formally began in 1909 when Alfred Weber considered how to locate a warehouse in such a way that the total distance between the warehouse and several customers was minimized [33]. Location theory once again attracted the interest of researchers in 1964 with a paper by Hakimi [34]. He wanted to locate switching centers in the communications network and police stations in the highway system. Research in the field of location can be divided into three parts: location problems, allocation problems, and location-allocation problems [33]. The main research content of this paper can be seen as a location-allocation problem, which has attracted much attention in the mathematical science of facility location in discrete and continuous optimization over the last four decades [33]. Location-allocation models are used to locate optimal facility locations and allocate demand points to each facility based on a measure such as shortest travel distance or time. It is often applied to the study of the allocation of urban public service facilities such as educational resources [35,36], medical facilities [37,38], bioenergy facilities [39], and collection and recycling facilities [40].

There are two classical location problems: the p-median [41] and the location set covering problems [42]. Median problems are considered to be the main topic of location-allocation problems. These problems try to find the median among some candidate points to minimize the sum of costs. In a coverage problem, a customer can be served by a facility only if the distance between the customer and the facility is equal to or less than a predetermined number. The siting of public cultural spaces studied in this paper is more like a combination of these two problems. The ultimate goal of the path-clustering algorithm is to minimize the sum of the path distance values. However, the setting of the

number of clusters of this algorithm is determined by the travel distance, which means that the final clustering result should ensure that the shortest path distance between the villagers and the public cultural space is less than the set travel distance value.

There are three solution methods for the location allocation problem: exact solutions, heuristic methods and metaheuristic methods. Heuristic methods such as greedy assignment [43,44] find near-optimal solutions by examining only a limited subset of potential combinations. In contrast, metaheuristic methods such as the genetic algorithm (GA) have a more diverse output due to the inclusion of random factors in the algorithm solution. Finally, exact solutions such as branch and bound algorithms find the optimal solution by systematically examining a large subset of feasible combinations of all possible facility locations and demand allocation options. Although the path-clustering algorithm written in this paper is a type of heuristic algorithm, it is still possible to output diverse results by changing the initial parameters.

There is already more mature software that provides location-allocation analysis tools. For example, in ArcGIS software (https://www.esri.com/en-us/arcgis/about-arcgis/overview, accessed on 7 November 2022), the location-allocation function is available in the network analysis module. It can be used both to minimize the overall distance between the demand point and the facility and to maximize the number of demand points covered within a certain distance. These functions are very easy to use and efficient. However, since the tools are highly integrated, it is difficult to make changes to the algorithms based on requirements. On the other hand, the path-clustering algorithm written in this paper is more flexible. It can not only output a variety of results to better provide a reference for rural public cultural space planning but also all the data in its computing process are easily available, which can provide more references for decision-making.

*1.3. Our Study*

In this study, we propose a siting analysis method for public cultural spaces based on a path-clustering algorithm and use Yumin Township in Jilin Province as a test case. When a public cultural space needs to serve multiple villages, the travel distance of villagers is used as the constraint factor. Then, the clustering algorithm is used to divide the villages within the township into groups. Villages within the same group share one public cultural space, which can avoid wasting resources while meeting villagers' needs for cultural space. The villages are clustered around public cultural spaces. Transportation joins these clusters into a cultural system of the township. The organization is a tree [45]. In the cluster analysis, considering the practical situation, people's travel distance cannot be simply represented by the Euclidean distance between two points. Therefore, we write the path-clustering algorithm that uses the shortest path distance between points as the unit of measure to complete the clustering of villages in our study.

This paper is organized as follows: Section 2 provides an interpretation of the concept of public cultural space and its specific meaning in the Chinese rural context; Section 3 mainly describes the clustering algorithm written using the shortest path distance as the metric distance; Section 4 focuses on how the algorithm is applied to the clustering of villages; Section 5 discusses the novelty of this study and its potential applications, as well as the limitations of this study and future research works. Finally, Section 6 summarizes the main conclusions of this paper.

## 2. The Concept of Rural Public Cultural Space

*2.1. Public Cultural Space*

Public cultural space is an integrated concept with theories related to "public space" and "cultural space", which involves multidisciplinary fields such as sociology and cultural studies [46]. It is a spiritual and material community that carries cultural activities. The "public space" is generally considered to be the middle ground between the state and society, the place where public cultural activities are consciously carried out by the citizen class. The concept of "cultural space" first appeared in the book *The production of space* [47]

by French scholar Lefebvre. In this book, he mentions that space is a product of society, a process of life production, which has the property of triple dialectical interaction of perception, conception and reproduction. Later, the concept of "cultural space" was further expanded to refer to a physical or social space. It encompasses a set of behaviors and patterns of life associated with a particular group of people who occupy this space [48]. Public cultural space, as an integrator of two concepts, was initially defined in an urban context. In the book *The fall of public man*, Sennett mentioned the description of public cultural space as "Urban amenities were diffused out from a small elite circle to a broader spectrum of society, so that even the laboring classes began to adopt some of the habits of sociability, such as promenades in parks, which were formerly the exclusive province of the elite, walking in their private gardens or 'giving' an evening at theater"[49]. Therefore, in the context of urban culture, some public cultural spaces, such as squares, theatres, museums and art galleries, are not only physical spaces or places but also contain the meaning of "public", "shared" and "cultural". It has the significance of redefining modern urban life and human relations.

*2.2. The Concept of Public Cultural Space in the Chinese Rural Context*

In the rural context, different scholars have tried to define public cultural space. Li [50] proposed that rural public cultural space is a complex that carries regional culture and has social, economic and productive characteristics. At the same time, it also has the function of cultural edification and is the main carrier of rural cultural heritage. According to Zheng [51], rural public cultural space is a kind of community with both cultural and spatial properties, which is a spiritual and material construction of human social activities. In this study, rural public cultural space refers to a holistic concept that includes cultural activities and cultural resources and is the spatial carrier in which rural cultural life occurs. As a spatial carrier, rural public cultural space mainly refers to the space where many people can participate in cultural activities and that is not privately owned. There are many aspects in the construction of public cultural spaces. This paper focuses on the location of public cultural space. In this issue, we need to consider the spatial properties of the public cultural space. It should be more public, a place where farmers can freely access, interact and participate in public affairs.

*2.3. Types of Rural Public Cultural Spaces*

In the existing studies on rural public cultural spaces, most scholars have classified public cultural spaces. For example, Chen [52] divided rural public cultural space into living culture public space, traditional culture ritual space and folk-art exhibition space. Chen et al. [53] divided rural public cultural spaces into traditional rural public cultural spaces and new rural public cultural spaces according to the order of historical development. Traditional rural public cultural spaces include ancestral shrines, theatres and other spatial places with traditional history and culture. The new public cultural spaces are generally government-led or villagers' self-organized spaces for holding modern public cultural activities. These include rural museums, cultural and sports rooms, electronic reading rooms and cultural squares with fitness and recreational functions. Cao [54] classified public cultural spaces into two types, "administratively embedded" and "endogenous to the village", according to the different formation mechanisms. Luo et al. [55] took Zhulian Village in Liuyang City as an example and classified public cultural spaces into three types: productive, living and organizational. Some scholars have also classified public cultural spaces by their spatial morphological characteristics. For example, Zhu et al. [56] classified public cultural space into point-like space, linear space and surface-like space based on spatial form.

In this paper, we focus on the siting of public cultural spaces with the ultimate goal of achieving a reasonable distribution of cultural resources and meeting the needs of villagers. As different types of public cultural spaces differ in their scope of services, the content to be considered when conducting siting analysis will be different. Therefore, in this

paper, public cultural spaces are divided into three types according to the difference in the service scope. They are "single-village decentralized", "single-village concentration" and "multi-village concentration" (Figure 1). The "multi-village concentration" type of public cultural space refers to one rural public cultural space serving multiple villages, such as libraries, cultural stations and other cultural and educational public cultural spaces. The "single-village concentration" type of public cultural space refers to one public cultural space serving one village, such as a small square or a cultural compound. The "single-village decentralized" type refers to multiple public cultural spaces serving one village. These are generally living public cultural spaces. Examples include the open space outside the courtyard, which is used for villagers to chat or for some simple productive activities. This paper focuses on the "multi-village concentration" type of public cultural space. Since this type of public cultural space serves a larger area, its spatial dimensions are generally larger and can accommodate complex activity functions. It is generally established by the government, with more resources and manpower invested, and forms faster. This kind of public cultural space carries the task of enriching the cultural life and improving the cultural level of rural residents. Therefore, good planning is needed to meet the villagers' needs for cultural space.

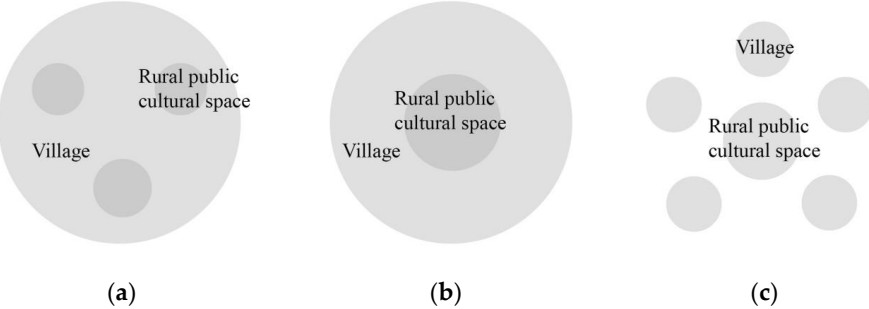

(**a**) (**b**) (**c**)

**Figure 1.** Schematic diagram of the relationship between three types of rural public cultural spaces and villages. (**a**) single-village decentralized; (**b**) single-village concentration; (**c**) multi-village concentration.

In constructing this type of public cultural space, it is necessary to divide the villages into clusters and to establish a public cultural space in each cluster. Therefore, how to divide the group and set up the public cultural space in which the village in the group is the focus of this study. We determine a constrained travel path distance based on the accessibility of the space of the actual path distance and the villagers' willingness to travel. Then, we divide all the villages in the township into clusters by the shortest path distance and determine the central villages for establishing public cultural spaces. The distance from the central village to any village in the group is guaranteed to be no greater than the set travel path distance.

## 3. Path-Clustering Algorithm

A clustering algorithm partitions a dataset into different classes or clusters according to specific criteria so that data objects within the same cluster are as similar as possible. It can also be understood as dividing data with similar attributes into one cluster, while data not in the same cluster are as different as possible. Typical cluster models include connectivity models, centroid models, distribution models, etc. In this study, the K-means [57] method is used as a partitioned clustering method. It requires specifying the number of clusters and the initial clustering centers in advance and dividing the data according to the similarity of eigenvalues by iterative iterations. In the K-means algorithm, the Euclidean distance is generally used as the similarity measure between clusters. However, the clustering classification based on Euclidean distance cannot meet the needs of this study. One cannot use the straight line distance from point to point as the basis of travel but instead the path

distance between points. Therefore, we need to write the clustering algorithm using the path distance as the similarity measure, which is the path-clustering algorithm.

### 3.1. The Application Tools Used in Writing the Path-Clustering Algorithm

The tools used to write the algorithm are Rhino, Grasshopper, GhPython and GH_CPython (Figure 2). First, we used Rhino and Grasshopper to build the path model. Then, the GhPython component in Grasshopper is used to calculate the shortest path distance between points and output the distance matrix. Finally, the clustering algorithm is written in the GH_CPython component to read the path distance matrix between points and output the clustering result of the point data. GhPython is a Python interpreter component for Grasshopper that allows us to run any type of dynamic script. Unlike other scripting components, GhPython supports rhinoscript syntax. This means that it can read the point and line data in the Rhino software (https://www.rhino3d.com/, accessed on 7 November 2022) so well that it is easy to determine if there is a colinear relationship between two points with a few simple codes. However, one disadvantage of this component is that it cannot call external python modules. Therefore, the path-clustering algorithm is mainly written in GH_CPython, which is also a component of Grasshopper and can call external python modules such as numpy. This means that GH_CPython can read the distance matrix very easily and apply it to cluster analysis. Finally, the clustering results are fed into Rhino and Grasshopper for visual display.

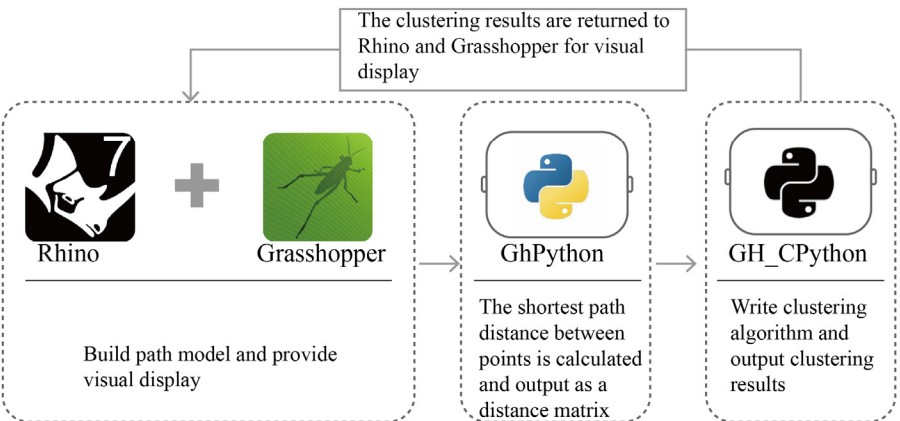

**Figure 2.** The application tools used in writing the path-clustering algorithm.

### 3.2. Process Overview of the Path-Clustering Algorithm

The general k-means clustering algorithm proceeds as follows: given a set of observations $(x_1, x_2, \ldots, x_n)$, where each observation is a d-dimensional real vector, k-means clustering aims to partition the $n$ observations into $k (\leq n)$ sets S = $\{S_1, S_2, \ldots, S_k\}$ to minimize the within-cluster sum of squares (WCSS) (i.e., variance). Formally, the objective is shown in Equation (1).

$$\arg\min_{S} \sum_{i=1}^{k} \sum_{x \in S_i} \| x, -, \mu_i \|^2 \tag{1}$$

where $\mu_i$ is the mean of points in $S_i$

$$\arg\min_{S} \sum_{i=1}^{k} \sum_{x \in S_i} \text{Dis}(x, \mu_i) \tag{2}$$

where $\mu_i$ is the point with the smallest sum of shortest path distances to all other points; Dis$(x, \mu_i)$ means the shortest path distance between $x$ and $\mu_i$.

However, in the path-clustering algorithm, the objective is to minimize the sum of the shortest path distances from points in different clusters to the centroid ($\mu_i$) (Equation (2)), where $\mu_i$ is the point in the cluster with the smallest sum of shortest path distances to all

other points. First, specify the value of K [58] and its initial cluster centroid, and then run the algorithm in its two steps alternately as follows:

Assignment step: The first step is to find the shortest path distance between each point and the initial cluster center and then assign each point to the cluster center closest to it. The clustering centers and the object points assigned to them form a cluster $S^{(t)}$ (Equation (3)).

Update step: For each cluster obtained in the previous step, find again the point that has the smallest sum of path distances to other points. This point is then used as the center of the clusters to redivide the new clusters, and the cycle continues in this way (Equation (4)). The algorithm is stopped when the centroids of the clusters are all coincident with the centroids of the previous step.

$$S_i^{(t)} = \left\{ x_p : \parallel x_p - m_i^{(t)} \parallel^2 \le \parallel x_p, -, m_j^{(t)} \parallel^2 \forall j, 1 \le j \le k \right\} \tag{3}$$

$$m_i^{(t+1)} = u_i^{(t)} \tag{4}$$

*3.3. The Shortest Path Distance between Points*

The shortest path problem is one of the classical problems in graph theory [59]. The goal of the shortest path problem is to find a path between two vertices (or nodes) in a graph that minimizes the sum of the weights of its constituent edges. Finding the shortest path between two points on a road map may be described as a specific example of the shortest path problem in graphs, where the vertices correspond to intersections and the edges correspond to road segments, each weighted by the segment length. In finding the shortest path distance between points, there are two main methods: the Dijkstra algorithm [60] and the Floyd algorithm [61]. This study uses a Floyd algorithm to find the shortest path distance between any two points. The algorithm logic is that for any pair of vertices U and V, see if there exists a vertex W such that the sum of the path distance from U to W plus W to V is shorter than the path distance directly from U to V. If so, update the shortest distance between the two points U and V. The logic of the algorithm is demonstrated in the path diagram shown in Figure 3. The shortest path problem can be defined for graphs that are undirected, directed or mixed [62]. In the study of this paper, the path graph can be considered an undirected graph. The weights of the edges in the graph are their lengths. First, the distance value from each point to all other points in the graph is calculated and saved in the form of a matrix. If two points are connected by a line, the length of the line is recorded as the distance value between the two points. If there is no line between the two points, the distance between the two points is recorded as infinity. Once the initial distance values from each point to all other points are available, the shortest path distance between the points can be calculated in a loop based on these values. First, point A is chosen as the middle point, which does not make the distance between the other two vertices shorter; then, the distance matrix is not updated. The distance matrix is updated when Dis[A][D] > Dis[A][B] + Dis[B][D], Dis[A][E] > Dis[A][B] + Dis[B][E], and Dis[C][E] > Dis[C][B] + Dis[B][E] with point B as the midpoint. When C is the midpoint, the distance matrix is also kept constant. When D is the midpoint, Dis[C][E] > Dis[C][D] + Dis[D][E], Dis[A][F] > Dis[A][D] + Dis[D][F], Dis[B][F] > Dis[B][D] + Dis[D][F] and Dis[C][F] > Dis[C][D] + Dis[D][F]. Update the distance matrix, and then follow this method to loop and continue to update the distance matrix with E and F as the midpoints. Finally, the distance matrix of the shortest path distance between all points is obtained (Figure 4).

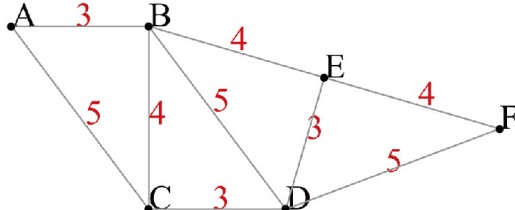

**Figure 3.** Path diagram-1. Letters A–F are the index value of each point. The numbers represent the length of each line.

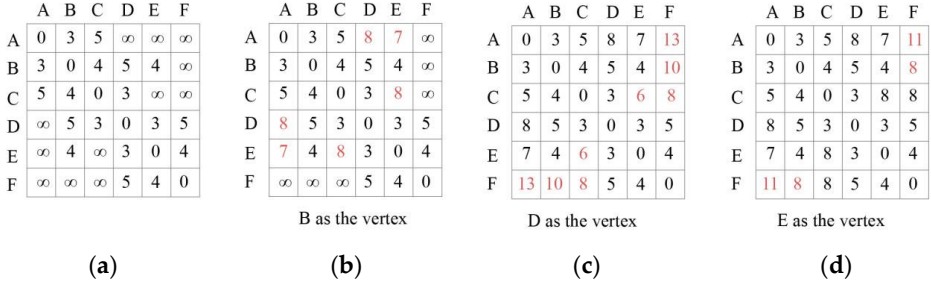

|   | A | B | C | D | E | F |
|---|---|---|---|---|---|---|
| A | 0 | 3 | 5 | ∞ | ∞ | ∞ |
| B | 3 | 0 | 4 | 5 | 4 | ∞ |
| C | 5 | 4 | 0 | 3 | ∞ | ∞ |
| D | ∞ | 5 | 3 | 0 | 3 | 5 |
| E | ∞ | 4 | ∞ | 3 | 0 | 4 |
| F | ∞ | ∞ | ∞ | 5 | 4 | 0 |

|   | A | B | C | D | E | F |
|---|---|---|---|---|---|---|
| A | 0 | 3 | 5 | 8 | 7 | ∞ |
| B | 3 | 0 | 4 | 5 | 4 | ∞ |
| C | 5 | 4 | 0 | 3 | 8 | ∞ |
| D | 8 | 5 | 3 | 0 | 3 | 5 |
| E | 7 | 4 | 8 | 3 | 0 | 4 |
| F | ∞ | ∞ | ∞ | 5 | 4 | 0 |

B as the vertex

|   | A | B | C | D | E | F |
|---|---|---|---|---|---|---|
| A | 0 | 3 | 5 | 8 | 7 | 13 |
| B | 3 | 0 | 4 | 5 | 4 | 10 |
| C | 5 | 4 | 0 | 3 | 6 | 8 |
| D | 8 | 5 | 3 | 0 | 3 | 5 |
| E | 7 | 4 | 6 | 3 | 0 | 4 |
| F | 13 | 10 | 8 | 5 | 4 | 0 |

D as the vertex

|   | A | B | C | D | E | F |
|---|---|---|---|---|---|---|
| A | 0 | 3 | 5 | 8 | 7 | 11 |
| B | 3 | 0 | 4 | 5 | 4 | 8 |
| C | 5 | 4 | 0 | 3 | 8 | 8 |
| D | 8 | 5 | 3 | 0 | 3 | 5 |
| E | 7 | 4 | 8 | 3 | 0 | 4 |
| F | 11 | 8 | 8 | 5 | 4 | 0 |

E as the vertex

| (**a**) | (**b**) | (**c**) | (**d**) |
|---|---|---|---|

**Figure 4.** Transformation of the distance matrix. (**a**) Initial distance matrix; (**b**) distance matrix after B as midpoint; (**c**) distance matrix after D as midpoint; (**d**) distance matrix after E as midpoint.

### 3.4. Initialization Methods

In the K-means clustering algorithm, the common initialization methods are Forgy and random partition [63]. The starting means in the Forgy technique are determined by randomly selecting k observations from the dataset. The random partition technique initially randomly allocates a cluster to each observation and then goes to the update stage. The initial mean is determined to be the centroid of the cluster's randomly allocated points. In this study, the centroids of the clusters are specified randomly in the initialization phase using the Forgy technique. Taking the path diagram in Figure 5 as an example. It is a model consisting of points and lines on Rhino, where points are directly connected to points on a straight line, and the values on the line indicate the length of the line. The ultimate goal of the path-clustering algorithm is to cluster the point data in the path graph, with the similarity measure being the shortest path distance from point to point. The algorithm first finds the shortest path distance between all points to obtain the path distance matrix and then performs the clustering analysis based on this distance matrix. When calculating the path distance between points and outputting the distance matrix. If the two points are on the same line, its shortest path distance is equal to the length of the line connecting the two points, such as point A and point B in Figure 5. If the two points are not on the same line, then the path distance between the two points needs to be calculated by the algorithm, such as point A and point C in Figure 5.

We choose the number of clusters as 5 and calculate the within-cluster sum of path-distance (WCSPD) between each result. The WCSPD value represents the sum of the distance values from the points in all clusters to the centroid of the clustering result (Equation (5)), and the smaller its value is, the better the clustering result. By comparing the WCSPD values, it is possible to compare the variability between the clustering results corresponding to different initialized clustering centers. As seen from the following nine clustering results (Figure 6), their clustering results are relatively close to each other in terms of WCSPD values. The results show that the more discrete the choice of their initial clustering centroids, the better their final results.

$$\text{WCSPD} = \sum_{i=1}^{k} \sum_{x \in S_i} \text{Dis}(x, \mu_i) \tag{5}$$

where $\mu_i$ is the point with the smallest sum of shortest path distances to all other points; $Dis(x,\mu_i)$ means the shortest path distance between $x$ and $\mu_i$.

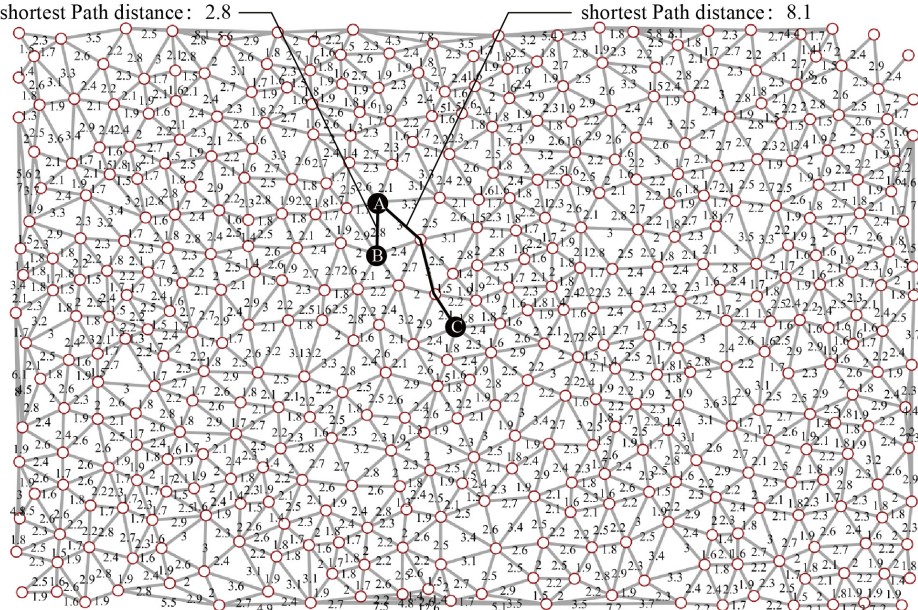

**Figure 5.** Path diagram-2. This is a model consisting of points and lines on Rhino, where points are directly connected to points by straight lines. The values on the line indicate the length of the line.

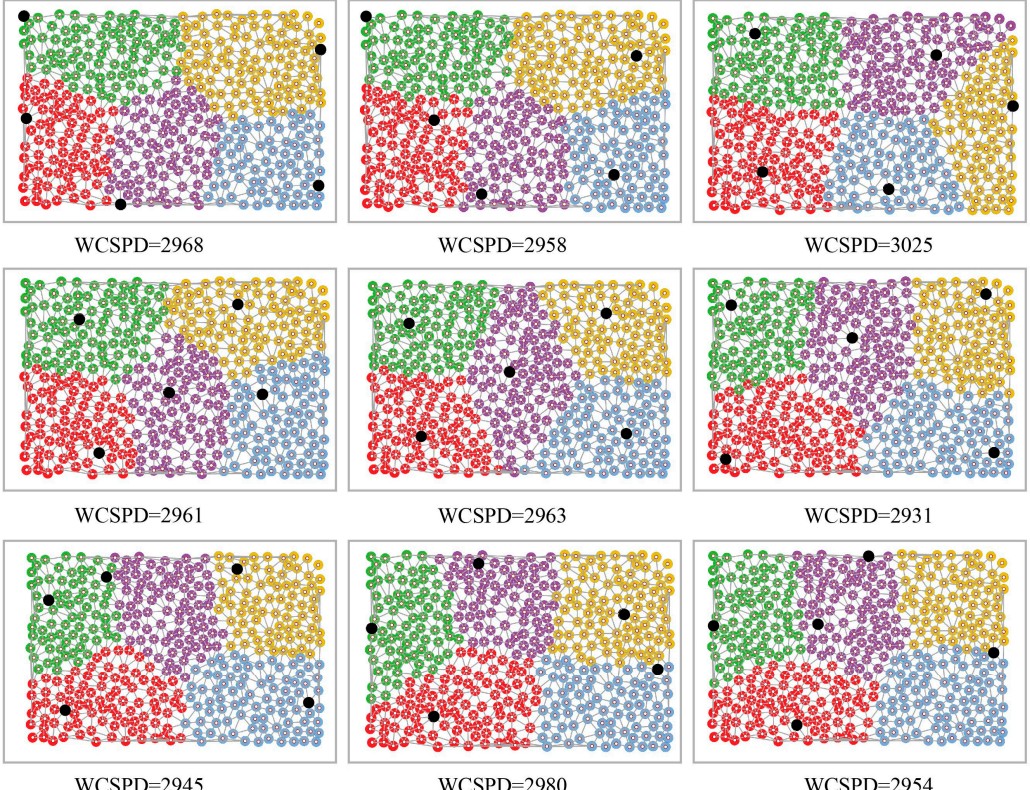

**Figure 6.** The WCSPD values are obtained from the clustering results corresponding to different initial clustering centers. The points of the same color in the graph belong to the same cluster. The black points are the initial clustering centers.

### 3.5. K Value: Number of Clusters

In this study, the results of clustering need to ensure that the maximum value of the path distance from the points in all clusters to the centroid does not exceed the set travel distance. In other words, the maximum value of the path distance within-cluster (MPDWC) needs to be less than the set value (Equation (6)). Therefore, the number of clusters is specified by the set travel path distance value. This value is influenced by a number of factors, including the mode of travel or the specific function of the destination to be reached. We can use the stated preference method [64,65] to obtain this value. Again, using the path diagram in Figure 5 as an example, the MPDWC values of the clustering results corresponding to different K values (number of clusters) in this path diagram are plotted as a line graph. From the results (Figure 7), it can be found that the MPDWC values decrease as the K values increase. Therefore, once we obtain the limited travel path distance value, we can choose a suitable K value.

$$\text{MPDWC} = \max_{i=1}^{k} \max_{x \in S_i} \text{Dis}(x, \mu_i) \tag{6}$$

where $\mu_i$ is the point with the smallest sum of shortest path distances to all other points; $\text{Dis}(x, \mu_i)$ means the shortest path distance between $x$ and $\mu_i$.

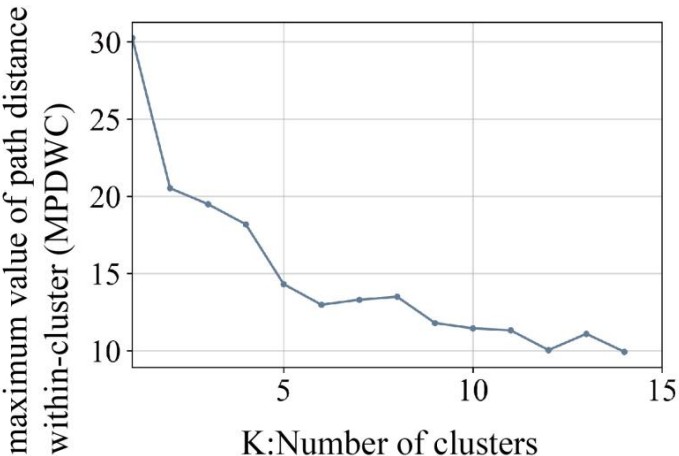

**Figure 7.** The line graph of the MPDWC values corresponding to different k values.

## 4. Applying the Path-Clustering Algorithm to the Siting Analysis of Rural Public Cultural Spaces in Yumin Township

### 4.1. Yumin Township

The topography of Jilin Province gradually rises from northwest to southeast. With Dahei Mountain in the central region as the boundary, it is mainly divided into two major landscapes: the mountains in the southeast and the plains in the central and western regions (Figure 8a). Villages in Jilin Province are more concentrated in the central region and less distributed in the southeast region (Figure 8b). The villages in Jilin Province can be divided into three regions based on the topography and the spatial density characteristics of village distribution. They are high-density plains areas, medium-density plains areas and low-density mountainous areas (Figure 9). Yumin Township is located in high-density plain areas. Since the clustering algorithm divides villages into groups and the distribution of villages within Yumin Township is relatively scattered (Figure 10), it is well suited as a research object. Yumin Township is located in the northern part of Yushu city, Jilin Province, in the hinterland of the Songliao Plain. It is adjacent to Daling Township in the east, Taian Township in the south, Hongxing Township in the west and Lalin River in the north, with a total area of approximately 151 square kilometers. The plan of Yumin Township (Figure 10) is approximately rectangular, with a length of approximately 17 km from north to south and a width of approximately 12 km from east to west. The villages within the township are relatively evenly distributed.

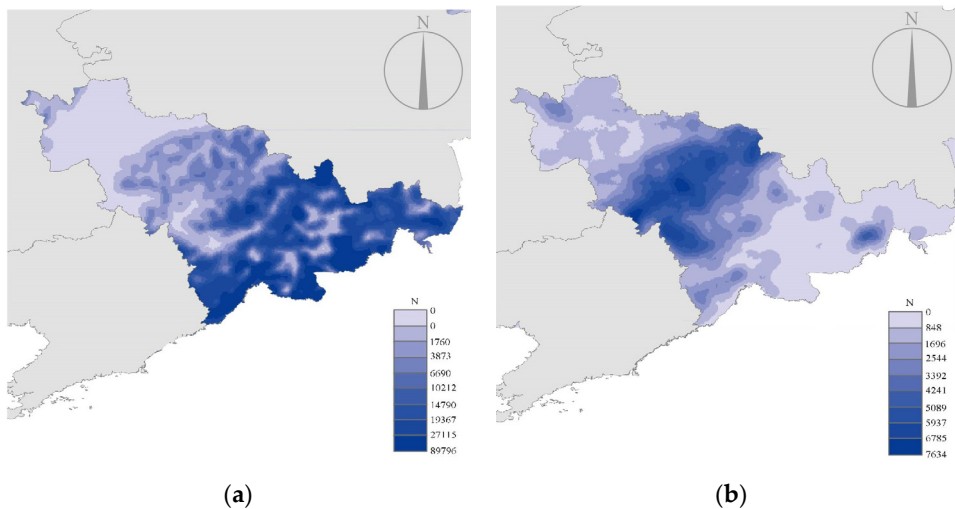

(**a**)                                          (**b**)

**Figure 8.** The kernel density analysis results. (**a**) the kernel density analysis of elevation data in Jilin; (**b**) the kernel density analysis of POI data of villages in Jilin.

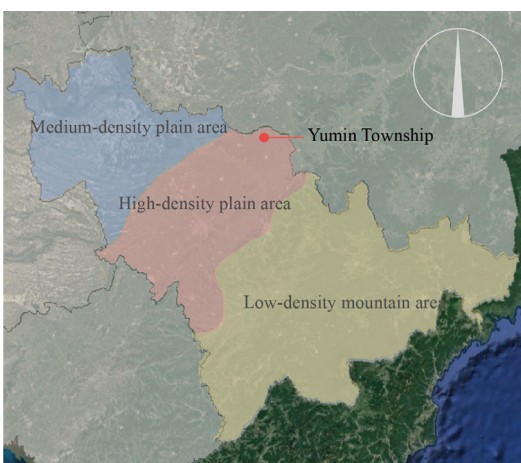

**Figure 9.** Schematic diagram of the three areas. Villages in Jilin Province are divided into three regions according to their distribution characteristics.

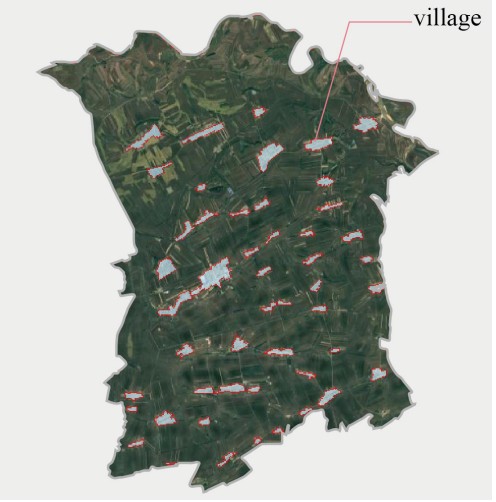

**Figure 10.** Satellite map of Yumin Township.

### 4.2. Clustering Analysis

Before the clustering analysis of villages in Yumin Township, the path diagram of the township needs to be manually depicted based on the satellite map (Figure 11). The elevation difference within the town area is within 30 m, which is negligible compared with the area of the township. Therefore, the elevation difference cannot be considered when drawing the path model. We know that there are different types of roads in the city, such as green streets, looped local roads and promenades [66]. However, there are basically no different road types in the study townships of this paper. Therefore, the road types were not classified when drawing the path diagram. Only the original path distances were retained.

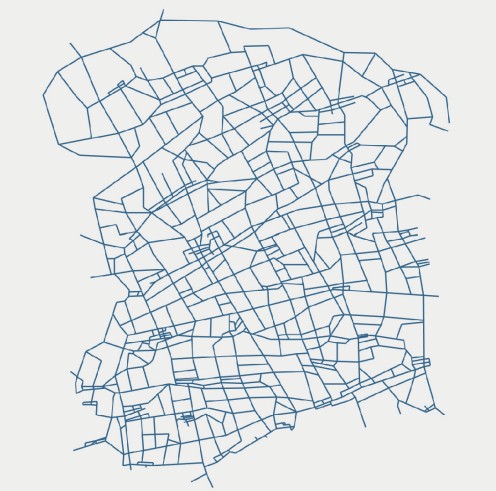

**Figure 11.** Path diagram of Yumin Township.

This path diagram is a line model on Rhino, which is needed for processing before starting the clustering analysis. First, we need to generate endpoints, intersections and turning points on the line model. All lines are disconnected by these points. Finally, we need to obtain a model consisting of points and lines, and the connecting lines between the points are straight lines (Figure 12). Then, the shortest path distance between points is calculated, and the distance matrix is output.

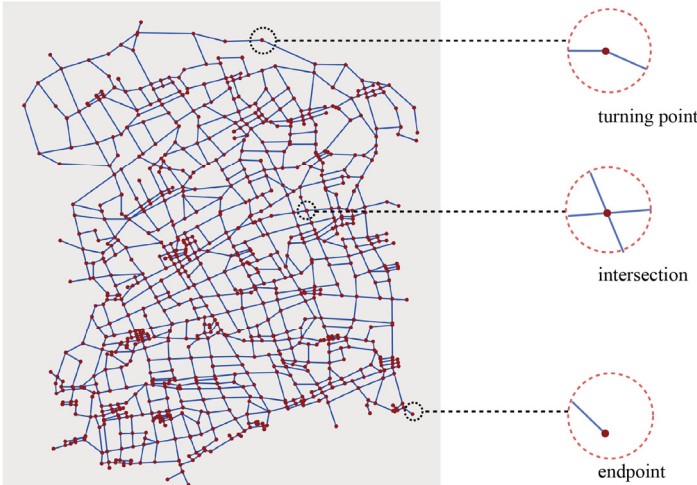

**Figure 12.** The path diagram of Yumin Township after processing. The model consists of points and lines on Rhino, where the connecting lines between the points are straight lines.

Since the ultimate goal of the algorithm is to cluster the villages, it is not necessary to perform cluster analysis on all the point data in the path map. Therefore, we can filter the

data of the points on the path and keep only the points that can represent the villages. The point with the smallest sum of path distances to other points in the village was selected as the representative (Figure 13), and a total of 44 points were selected to represent 44 villages (Figure 14). Therefore, the distance matrix obtained above needs to be filtered to obtain the shortest path distance matrix between each point of the village in preparation for the cluster analysis. The original is a 969 × 969 matrix representing the shortest path distance values between 969 points. The converted one is a 44 × 44 matrix representing the shortest path distance value between 44 points.

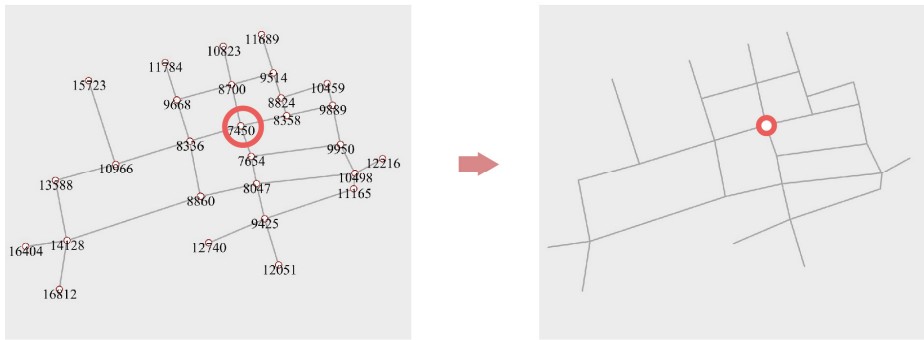

**Figure 13.** This picture shows how the point data are filtered. The point with the smallest sum of path distances to other points in the village was selected as the representative. The value on each point on the left graph represents the sum of the shortest path distances from that point to all other points.

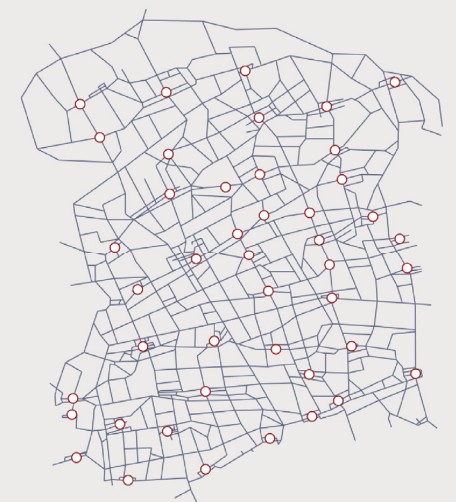

**Figure 14.** Path diagram of Yumin Township after filtering the point data. The 44 points in the figure represent 44 villages.

First, a line graph of the variation in MPDWC values with K values is plotted (Figure 15). Then, the K values are selected according to the set shortest path distance for the trip. As mentioned before, there are more things to consider when determining the path distance of the trip, and the stated preference method can be used to obtain this value. In this paper, we choose two values randomly to show the results of this method. In the analysis of village clustering in Yumin Township, we chose two different shortest path distances for travel, which were 5 km and 6 km. From Figure 15, we can see that when the travel path distance is set to 5 km, the K value is set to eight. When the travel path distance is set to 6 km, the K value is set to six.

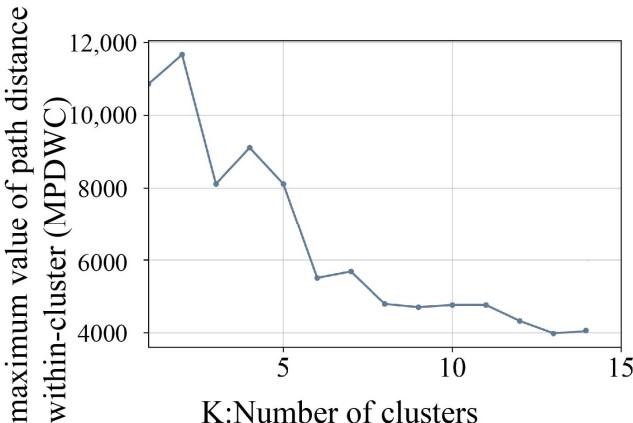

**Figure 15.** The line graph of the MPDWC values corresponding to different k values for the results of the cluster analysis of the path map of Yumin Township.

### 4.3. Clustering Results

Figure 16 shows the clustering results corresponding to three different initial clustering centers with k values of eight. From the clustering results, we can see that the villages in Yumin Township are divided into eight groups, which means that eight multi-village concentration-type public cultural spaces need to be set up. The number of villages in each cluster varies from a maximum of eight to a minimum of three. Therefore, the scale of the public cultural space they need to set up varies and needs to be set specifically according to the number of villages they serve. The three clustering results also have different MPDWC values, but the differences are not very large, and all are less than the set travel distance of 5 km. Figure 17 shows the clustering results corresponding to three different initial clustering centers with k values of six. The villages in Yumin Township are divided into six clusters, which means that six multi-village concentration-type public cultural spaces need to be set up. The number of villages in each cluster has a maximum of eleven and a minimum of three. Notably, the output results are diverse in the initial stage of the algorithm due to the different initial clustering centers specified. Therefore, there may also be a situation where the MPDWC value of the clustering result may exceed 6 km. The initial clustering center needs to be reselected when this situation is encountered, and then the clustering results are filtered to meet the requirements.

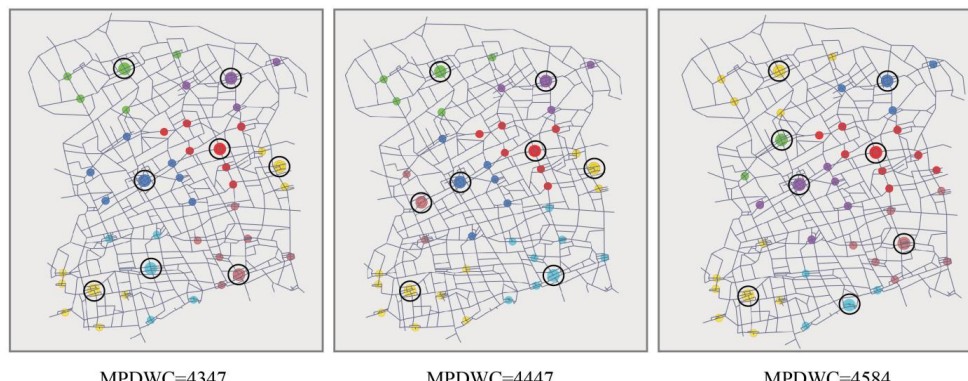

| MPDWC=4347 | MPDWC=4447 | MPDWC=4584 |

**Figure 16.** The clustering results and MPDWC values when the value of K is eight. The points circled by black circles are the centroids of each cluster in the clustering result.

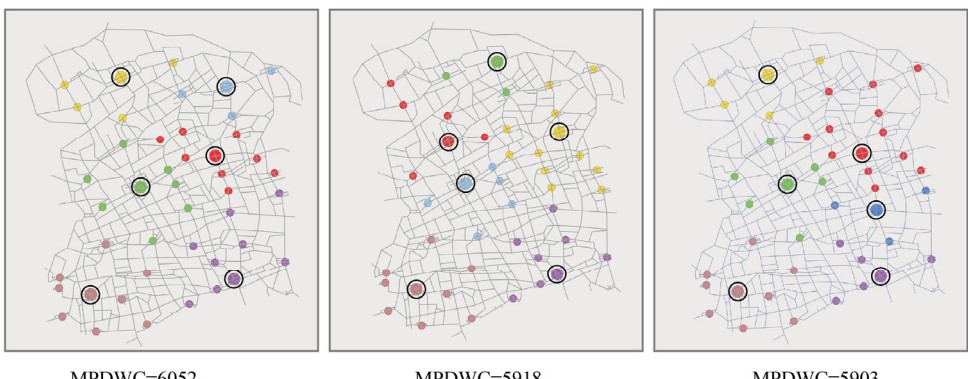

**Figure 17.** The clustering results and MPDWC values when the value of K is six. The points circled by black circles are the centroids of each cluster in the clustering result.

In summary, there is variability in the clustering results for different initial clustering centers. The final purpose of cluster analysis is to divide villages into groups, and the villages corresponding to the final cluster centroids are used as villages for establishing public cultural spaces. Therefore, the goodness of its clustering result is not only judged by the WCSPD or MPDWC values but also depends on whether the village of the center point of the final clustering is suitable as a village for setting up public cultural space. If its scale is too small, it is not suitable as a village for setting up public cultural space. Therefore, multiple clustering results can be generated based on different initial clustering centers, and then the appropriate centers can be filtered according to the needs.

## 5. Discussion

Against the background of rural revitalization, an increasing number of scholars have started to engage in research on rural construction. Rural cultural revitalization is an essential part of rural revitalization. Therefore, as a carrier of rural culture, rural public cultural space has received increasing attention. However, current research on this topic is mostly theoretical and lacks support from scientific and quantitative research methods. In contrast, in studies of the spatial distribution of some types of public service facilities in cities, many scholars have adopted quantitative research methods. The purpose is to rationalize and optimize the spatial distribution of public services and to use it to solve the existing problems. In the study of this paper, we also hope to apply scientific and quantitative research methods to study the problems related to rural public cultural spaces. Therefore, based on the characteristics of rural public cultural spaces, a path-clustering algorithm is written, and then the algorithm is applied to study the location problem of rural public cultural spaces. Many studies apply clustering algorithms to spatial distribution. However, few clustering algorithms have been written using the actual path distance as the similarity metric, mostly based on Euclidean distance. Therefore, the study is not only more relevant to reality, in line with the law of human travel but also has practical application value.

Many existing GIS software programs have this function and may be more efficient due to their greater maturity. However, these are highly integrated and more like black boxes that cannot be adjusted to our needs. In contrast, the path-clustering algorithm written in this paper is more flexible. It can not only directly specify the number of public facilities for analysis but also obtain the number of facilities based on the set travel distance before analysis. It can output multiple results to better provide a reference for rural public cultural space planning. Moreover, all the data in its computing process are easily available, which can provide more references for decision-making.

In this paper, we chose Yumin Township as the test object of the method, whose villages are scattered within the township and have a small overall height difference. Therefore, only the path distance on the plane was considered when conducting the analysis. This method

is still applicable to townships with large height differences, and only a three-dimensional road network needs to be constructed. In general, this algorithm is better applied in townships where villages are scattered and somewhat less applicable in townships where villages are all close together. This siting analysis method for public cultural spaces can be applied not only to townships such as Yumin, which lack public cultural spaces, to provide a reference for the planning of new public cultural spaces, but also to villages where public cultural spaces already exist to provide suggestions for optimizing the distribution of their current public cultural spaces. The analysis of the current distribution of public cultural spaces leads to the conclusion that the number of cultural spaces or the location of public cultural spaces should be changed to make the distribution of resources more reasonable.

The following two problems still exist in the current study. One is that the algorithm is time-consuming. Taking the path map of Yumin Township studied in this paper as an example, it takes 1.2 h to complete the clustering analysis. Second, the siting study of this public cultural space takes into account the travel distance of people. However, it has not been tested in practical projects to determine the effectiveness of its application. For example, we ignore the differences between paths when we build the model, while in reality, we know that the variability between different paths can have an impact on human choices. Therefore, future research should focus on two aspects. One is to optimize the algorithm to improve the speed and accuracy of this algorithm. The algorithm can be mainly divided into two parts. One is to establish the shortest path distance matrix between points, and the other is to perform cluster analysis based on the distance matrix. The most time-consuming part is to build the distance matrix. Therefore, our next work will be optimized for this part. This includes choosing a more efficient algorithm to determine the distance matrix between points. We also need to further investigate the influence of the hyperparameters on the clustering results and how to set the hyperparameters to make the algorithm more efficient. Second, the research will be integrated with practical applications, and the results of the practical applications will be used to evaluate the effectiveness of the research. It is also worth noting that the practical applications of this study go far beyond this. In the future, we will try to apply this algorithm to other spatial distribution studies of public spaces to further broaden the application dimension of this study.

## 6. Conclusions

A path-clustering algorithm was written according to the logic of the k-means algorithm in this study. Then, the clustering algorithm was applied to study the siting of rural public cultural spaces. In Yumin Township, villages within the township are divided into clusters based on the distance people can travel. In each grouping, a central village is selected to set up a rural public cultural space to serve all villages. On the basis of the high accessibility of rural public cultural space, the distribution of its public cultural resources is also more even. It not only meets the villagers' demand for space but also saves public cultural resources to a certain extent. This study also has the following findings.

(1) Its initial hyperparameters have an impact on the clustering results. Among them, the initial clustering center affects the clustering results, but its WCSPD value does not change much. The K-values need to be selected according to the set travel path distance values. The MPDWC corresponding to different K values is first calculated, and then the K value is determined according to the travel path distance.

(2) Although the efficiency of this analysis method is not yet high compared to existing location allocation models, it is more flexible and allows the algorithm to be adjusted based on individual needs.

(3) The analysis results are diverse, and different clustering results can be obtained by modifying the parameters, which can be filtered according to our needs. The final analysis results are only used as a reference, and more objective factors must be considered in the construction of cultural space. Nevertheless, the application of this quantitative analysis method to the study of the construction of rural public cultural space is very valuable and has certain practical significance.

**Author Contributions:** Conceptualization, D.L. and K.W.; methodology, D.L. and K.W.; writing algorithms, D.L.; validation, D.L. and K.W.; formal analysis, D.L.; writing—original draft preparation, D.L.; writing—review and editing, D.L.; supervision, K.W. All authors have read and agreed to the published version of the manuscript.

**Funding:** This research received no external funding.

**Institutional Review Board Statement:** Not applicable.

**Informed Consent Statement:** Not applicable.

**Data Availability Statement:** Not applicable.

**Conflicts of Interest:** The authors declare no conflict of interest.

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
