# Peer review of "Research on the Siting of Rural Public Cultural Space Based on the Path-Clustering Algorithm: A Case Study of Yumin Township, Yushu City, Jilin Province, China"

_sustainability, doi:10.3390/su15031999_

Round 1
Reviewer 1 Report
thank you for the opportunity to review this paper
the authors have conceived their activity on the creation of an algorithm that already exists in the most used GIS software
Nothing says about the criteria for choosing the villages considered to host the centers
Nothing say about the choice and classification of paths, distances considered and means of transport
Other notes are included in pdf attached

Reviewer 2 Report
This is an interesting paper that developed a clustering algorithm to locate suitable public cultural spaces in rural areas. This is a GIS task to locate facilities. The paper seems to lack a discussion to compare the proposed method with others. With the slow speed in clustering, it would be great to explore whether existing methods can help to make the whole algorithm more efficient.
Here are some detailed comments:
1. In the abstract, lines 21-22, it is quite confusing to find some empirical results as the contribution of the method paper.
2. In the 1.3 case study, why Yumin Township is suitable to test the method? Different rural settings will require customized or tuning of the algorithm. Please add more reasons if the method is generalizable.
3. The overall literature review shall be enriched with a comparison with existing GIS-based facility allocation methods.
4. The concept of rural public cultural space is not so relevant to the method. This can be shortened giving more chances to discuss the methods.
5. Page 5, there is a lack of discussion about the reason you classify the phenomenon differently. Given the multi-village concentration as the focus, the reason for choosing it needs more elaboration. Whether the method is generalizable to a broader context may also relate to this focus.
6. Page 6, line 199, why use centroids? Any particular reason?
7. Figure 4,5,6 shall be explained in more detail. What are the points and lines? At the moment they are out of context. As indicated by Figure 6, MPDWC shall be included in the method. As the method develops, there could be an indication of a best option of K given the long tail shown in Figure 6.
8. Figure 8, why paths are derived from an image? Is any network data available?
9. In 4.2, you mention points data, folding, and turning points, in the path map, there is no initial points shown in Figure 8. A clarification of what are the points and how you defined them is needed.
10. Also in 4.2, why filtering is necessary? Why direct construction from the villages using the network is not ideal?
11. Figure 10 is confusing. What do the labels mean?
12. Page 13, line 326. You mentioned ‘There are clustering results whose MPDWC will exceed 6 km’. Isn’t it contradicting what the algorithm does? The method shall ensure the clusters are all within the threshold.
13. Page 14, it would be worthwhile to compare your algorithm with existing facility location models to optimize your method.
14. Page 14, there are many other forms of rural settings, and it needs to be discussed how and why this method can be generalized.
15. Conclusion (2), it is confusing to include empirical results as a method contribution. Please explain or modify.
Reviewer 3 Report
The article submitted for review raises very interesting issues of the need to carry out cultural revitalization as the basic form of rural revitalization.
The strength of the submitted publication is its methodological side. The authors created their own procedure and used the existing research methods to determine the number and distribution of the necessary cultural centers as an element of village revitalization.
Unfortunately, the article is very empirical and unilaterally spatial in nature. The publication lacks reference to this type of research from other parts of the world than China. Much has been written about the location of public institutions, including cultural centers (see: Christiaanse 2020 Neumeier, 2016; Westlund & Pichler, 2012; Brereton et al, 2011; Haugen et al, 2012; Meijers, 2007; Tacoli, 1998; Smoyer -Tomic, Spence & Amrhein, 2006; Woods, 2005). The article lacks considerations on the subject of the researched phenomenon inscribed in the issue of sustainable development. It is not entirely clear what new research brings to the current state of knowledge and what implications and recommendations follow from them.
The article submitted for review has great cognitive potential. However, the lack of embedding it in world literature and concepts at the present stage classifies it as a major revision.
Author Response
First, thank you very much for your review comments. This allowed us to clearly recognize the shortcomings of our paper and make changes.
We have rewritten this section of 1.2 related work and have included a large amount of world literature to enhance the context and meaning of this paper. This includes a review of existing research on public space in rural areas, the main research directions on the distribution of public facilities, and the application of location distribution models.
In this paper, we focus on proposing a site analysis method for public cultural spaces. This analysis is not applied to traditional cultural spaces with historical value but rather to new cultural facilities, such as libraries, that are built to improve the cultural level of farmers. Provide technical support for the planning of these cultural spaces. This siting analysis method takes the accessibility of space as the starting point of the study and the travel distance of villagers as the main influencing factor. It is able to output a variety of different results to provide a reference for public cultural space planning. There are some shortcomings in the current study, which we also point out in the discussion section of this paper. Since the analysis method still lacks a test of the practical application effect, the overall results seem to be somewhat imperfect. We will continue our research to solve this problem in future work. However, the method itself is now better described in this paper. We have followed your advice and added a large number of references to make the paper more complete. Thank you very much for your hard work!
Round 2
Reviewer 1 Report
Thank you for your work and the improvement of the paper.
I'm not yet sure about the necessity of a new algorithm.
You cited ArchiGIS as example of GIS software that it is not able to do what you want.
Are you sure that this software exists? I didn’t find any reference about it on web.
I Know very well ArcGIS by ESRI software: it is possible to personalize it and do what you write.
In my opinion this paper needs a more consistent practical application to the case study to be published. The authors suggest it for the future, I think it is necessary for this paper.
Author Response
Dear reviewer,
Thank you for your hard work. We greatly respect the peer review comments. However, we have a different opinion on your review comments. The specific reasons are as follows:
First, We were a little confused about your review comments. We had trouble understanding what you were trying to say. For example, the third sentence of your comment was "You cited ArchiGIS as example of GIS software that it is not able to do what you want". What is the meaning of this sentence. It does not have much connection with the meaning of the two sentences before and after it. The fourth sentence of your comment was “Are you sure that this software exists? I didn’t find any reference about it on web”. This statement is very confusing to us. We mentioned a lot of software in the article and which software you were referring to. We do not know if you have read our paper seriously before giving your review comments.
Second, we think there are some unspecific and subjective comments in your review. The fifth sentence of your comment was “I Know very well ArcGIS by ESRI software: it is possible to personalize it and do what you write”. you simply said that you knew the ArcGIS software so well that you thought it would meet our needs. What is the evaluation criteria for this level of knowing it very well. According to your meaning, can we reply that we also know the ArcGIS software very well so we think this software cannot meet our needs. In contrast, reviewer 2 will give specific comments and criteria in the review comments. For example, his (or her) comment on the part of our paper about the comparison between the algorithm we wrote and the GIS software was “Many GIS tools can be combined to bring better results, such as service area and location-allocation analysis together can consider both travel distance and the number of facilities”. We are very happy to accept such review comments and revise our paper. It will make our paper better.
Finally, you said, “In my opinion this paper needs a more consistent practical application to the case study to be published. The authors suggest it for the future, I think it is necessary for this paper”. We already explained to you why we did this in our last round of responses. We mentioned that this is a series of works. This current article is the beginning of this new research topic for us. In this paper, we focus on the siting analysis method for public cultural spaces. We will apply the method to practical research in our subsequent work. We will cite this article directly in a subsequent related paper so that we can analyze the practical case with more words. However, you were not able to accept our reasons and did not give any specific reasons.
In conclusion, we are somewhat disappointed with your review comments. We believe that you did not provide useful comments on this paper but rather evaluated our article from your personal preference. We think it is unfair to do so as a reviewer. Peer review is very important work, and it serves as a platform for researchers in the same field to communicate. Therefore, both authors and reviewers should respect the peer-review process. Finally, we respect your comments that do not approve of our work, and we appreciate your comments on our work. Of course, through your comments we believe that you have done a lot of outstanding research in related fields and we would appreciate it if you could share it for us to learn from.
Reviewer 2 Report
Thanks very much for the improved submission!
This version now addressed most of the issues that are previously raised.
It has more details about the method with examples.
It however still has some issues that may need to be explained more.
(a) Many Python libraries can calculate the shortest path. Why use the proposed combination if the performance is constrained?
(b) You claimed one of the contributions of this method is an independent specification of K. The example on page 9, still uses a pre-defined 5. It is less convincing to claim this as a methodological benefit.
(c) Many GIS tools can be combined to bring better results, such as service area and location-allocation analysis together can consider both travel distance and the number of facilities. They also can be run quickly to adjust the results for the different numbers of k. The benefits of this method may need more clarification.
(d) Figure 15 is not necessary. Based on the updated explanation of the method, it relates to network analysis or graph theory. Any comparisons to that?
Reviewer 3 Report
The article has been corrected as suggested. It is ready for publication in its present form.
Author Response
Dear Reviewer:
We really appreciate your approval of our work and thank you for all your hard work.
Best wishes.